# A Deep Learning-Based Encrypted VPN Traffic Classification Method Using Packet Block Image

**Weishi Sun, Yaning Zhang, Jie Li, Chenxing Sun and Shuzhuang Zhang ***

School of Computer Science, Beijing University of Posts and Telecommunications, Beijing 100083, China
* Correspondence: zhangshuzhuang@bupt.edu.cn

**Abstract:** Network traffic classification has great significance for network security, network management and other fields. However, in recent years, the use of VPN and TLS encryption had presented network traffic classification with new challenges. Due to the great performances of deep learning in image recognition, many solutions have focused on the deep learning-based method and achieved positive results. A traffic classification method based on deep learning is provided in this paper, where the concept of Packet Block is proposed, which is the aggregation of continuous packets in the same direction. The features of Packet Block are extracted from network traffic, and then transformed into images. Finally, convolutional neural networks are used to identify the application type of network traffic. The experiment is conducted using captured OpenVPN dataset and public ISCX-Tor dataset. The results shows that the accuracy is 97.20% in OpenVPN dataset and 93.31% in ISCX-Tor dataset, which is higher than the state-of-the-art methods. This suggests that our approach has the ability to meet the challenges of VPN and TLS encryption.

**Keywords:** deep learning; VPN traffic classification; image recognize

## 1. Introduction

In recent years, the study of network traffic classification has become a popular research topic [1–3]. It plays an important role in achieving better quality of service (QoS), network security, and network monitoring [4,5]. On the one hand, traffic classification can optimize network resource allocation for better QoS. For example, the network administrators of enterprises or campuses can observe the distribution of network traffic through network traffic classification, and then formulate some new strategies to improve the network efficiency of resource utilization [6]. On the other hand, in terms of network security, traffic classification is usually the first step in some network monitoring activities such as malicious detection [7]. Therefore, the network traffic classification has always been an indispensable study in network management and supervision.

Virtual private network (VPN) is a technology that establishes a private network on a public network to access Intranet resources remotely. It is simple to deploy and use. According to the statistics of the 45 largest VPN apps in Google Play and IOS App Store, VPN downloads have exceeded 134 million times all over the world [8], which means that using the VPN can be seen everywhere in the network. In addition, with increasing awareness surrounding users' privacy protection [9], encryption technology has seen rapid growth and is used more and more widely. Many VPNs use TLS encryption to enhance communication security and protect users' privacy. The use of VPN, especially VPN in a TLS tunnel, poses a great challenge to traffic classification. VPN encapsulates the original traffic and hides some information of the original message. Furthermore, if VPN uses TLS tunnels for encryption, TLS tunnels will further group and randomize the payload of traffic. This can result in the traffic payload hardly contributing to VPN traffic classification. Therefore, VPN and TLS encryption will affect some traditional traffic classification technologies, such as deep packet inspection (DPI) [10] and port detection [11].

At present, VPN traffic classification methods mainly include the fingerprint-based method, payload-based method and statistical feature-based method. Fingerprint-based method usually matches the traffic fingerprint to be detected with the fingerprint database to identify the traffic. Furthermore, it has high accuracy and is usually applied to some fine-grained traffic identification problems, such as Tor traffic identification [12]. The payload-based method directly imports all or part of the packet payload into a classification model. One representative of them is an end-to-end encrypted traffic classification method proposed by Wang et al. [13]. A payload-based method which usually adopts deep learning algorithm has the advantage of a simple feature extraction process without manual design. However, the payload-based method is affected by encryption and has a better effect on unencrypted VPN traffic or traffic encrypted in a specific way. The statistical feature-based method extracts statistical features from VPN traffic and then constructs a machine learning or deep learning model to classify these traffic. Statistical features which are not affected by encryption can effectively solve the problem of VPN traffic classification. Shapira et al. [14] used packet size distribution at a different time to create an image, which they called FlowPic. After that, these FlowPics were sent into a traditional CNN model (Lenet-5) for classification, and the final experiment achieved good results. However, the FlowPic method also has limitations, that is, feature collisions caused by single packet size distribution will affect the accuracy of VPN traffic classification.

Recently, many deep learning-based methods were proposed and achieved some good performance. However, most of them only focused on the encapsulation of VPN. When these VPN traffics used TLS encryption, the methods will be influenced. For example, Shapira et al. [14] achieved better classification effectiveness in an unencrypted ISCX-VPN dataset than in the encrypted ISCX-Tor dataset. This is because encryption will hide some traffic plaintexts and randomize the payload. In this regard, we proposed a Packet Block-based method in order to improve the effectiveness of encrypted VPN traffic classification. In this method, continuous packets in the same direction were aggregated as the basic units, which are called Packet Block. Thus, traffic can be represented as a Packet Block sequence. The Packet Block features use the data size distribution of the whole flow and some packet interaction relations between the communication parties to reduce not only the effect of TLS and VPN encapsulation but also the feature collision of different traffic types. Then, an image was created using Packet Block features, which were subsequently imported into a CNN model to train the classifier. Finally, our method achieved satisfactory classification accuracy. In this paper, the ISCX-Tor dataset posted online and the OpenVPN dataset captured by ourselves were used. The ACC of traffic classification problems under the ISCX-Tor dataset was 93.31%, and that under the OpenVPN dataset was 97.20%. It can be seen that our method achieved a good classification effect in both authoritative public datasets and self-captured datasets.

The rest of this paper is structured as follows: Section 2 describes related work. Section 3 mainly introduces the encrypted VPN network traffic classification method based on Packet Block image, Section 4 mainly introduces the experiment designs and results. Section 5 is the conclusion of this paper.

## 2. Related Work

This paper mainly studies the classification of VPN traffic in TLS tunnels. Current studies rarely discuss the dual effect of VPN encapsulation and TLS encryption on traffic classification. However, VPN traffic in the TLS tunnel has the dual characteristics of VPN traffic and TLS encryption. Therefore, the classification of VPN traffic can also bring enlightenment to the classification of VPN traffic in TLS tunnels explored in this paper. Recently, VPN traffic classification mainly uses fingerprint-based methods, payload-based methods and statistics-based methods.

Fingerprint-based method: in the handshake process of TLS encryption, the messages before the two programs exchange cipher suites are unencrypted, including the information in the Client Hello phase or Server Hello phase, and the certificate-related information

provided by the server. They can usually be used to construct the fingerprint characteristics of traffic. In addition, the fingerprint characteristics may also include DNS context information and the HTTP context information of traffic. Fingerprint-based methods usually extract fingerprint information from a large number of traffic to build a fingerprint database. Then, whenever a new traffic to be classified is encountered, its fingerprint is matched with the fingerprint in the database to obtain the final classification result. Johanna et al. [12] found that the issuer and subject fields in normal certificates often do not contain information such as location or company name, and use random generic names, which are formatted as www.+ base-32 code of 8–20 letters + Com or Net, so Johanna et al. [10] successfully identified tor traffic from the dataset containing tor traffic using the issuer and subject fields in the certificate as fingerprints. Fingerprint-based methods usually perform well in some fine-grained classification problems, such as website recognition.

Payload-based method: a payload-based method takes all or part of the packet payload as the input of the classifier. Wang et al. [13] proposed an end-to-end encrypted traffic classification method for the first time. This method uses the first 784 bytes of TCP payload as input data to build a 1D-CNN model for traffic classification. After that, many researchers proposed improved methods on the basis of Wang et al. The research of He et al. [15] aimed to convert the first few non-zero payloads of the session into gray images, and used a convolutional neural network (CNN) to classify the converted gray images. Finally, satisfactory results were achieved in both non-VPN traffic and VPN traffic. All of these methods used the packet payload as the feature. Furthermore, a deep learning model was introduced to build the classifier, which achieved good results. However, the method based on the payload characteristics will be affected by encryption. Wang et al. [13] succeeded in classifying the VPN traffic because their training dataset and test dataset used the same encryption method and encryption key. Bu et al. [16] also proposed that, for VPN traffic, only using the header of the packet will have a better classification effect than using the entire packet. Therefore, when TLS encryption is applied, the packet payload is no longer meaningful due to symmetric encryption, and the effect of the payload-based method will be greatly reduced.

Statistical feature-based method: statistical feature-based method extracts statistical features from packets and imports them into classifiers. Statistical features are often not limited by TLS encryption or VPN. Mohammad et al. [17] proposed a deep learning-based approach called "Deep Packet". Furthermore, their approach achieved good effectiveness in both the application identification task and traffic categorization task. Gil et al. [18] used time-related statistical features to implement the tor traffic classification. They applied C4.5, KNN and random forest as classifiers. Finally, the accuracy in the ISCX-Tor dataset is more than 80%. Iliyasu et al. [19] proposed a semi-supervised learning method based on a deep convolutional generative adversarial network (DCGAN), the basic idea of which is to use the samples generated by the DCGAN generator and unlabeled data to improve the performance of the classifier trained with a small number of labeled samples. Their approach achieved good accuracy on both QUIC and ISCX-VPN datasets. Qin et al. [20] calculated the payload size distribution probability (they called it PSD) of packets in a two-way flow. Then, they used Renyi cross entropy to identify the similarity between the PSD of the traffic to be detected and the specific application. Finally, they solved the classification problem of eight kinds of VPN applications. Shapira et al. [14] converted traffic into images for identification. This method used the packet size and packet arrival time to create images called FlowPic. Then, these FlowPics were sent to a CNN for classification. The classifier has excellent classification accuracy in traffic classification and application recognition. Although FlowPic uses the packet size and arrival time as features, the time-related features are greatly affected by the network environment and make little contribution to VPN traffic classification. Therefore, FlowPic actually takes the packet size as the dominant feature. However, the packet size is highly correlated with the protocol and traffic size. When including many traffic categories, they may have similar packet size distribution. Shapira et al. [14] also mentioned that, in their experiment, 70% of the VoIP traffic was

finally identified as file transmission, which was caused by the feature collision of packet size features.

For the classification of VPN traffic in the TLS tunnel in this paper, VPN encapsulation and TLS encryption will have a certain impact on the above methods. Fingerprint-based methods usually extract fingerprint information from the plaintext part of traffic, but there is no meaningful plaintext part of VPN traffic in a TLS tunnel. Thus, fingerprint-based methods will be difficult to apply to the classification of VPN traffic in TLS tunnel. Under the effect of TLS encryption, the payload-based method can hardly contribute to the classification of VPN traffic in the TLS tunnel. As for the statistical-feature-based method, because some statistical features are not affected by VPN encapsulation and TLS encryption, it can be a feasible solution to the classification of VPN traffic in the TLS tunnel. This paper proposes a method based on the Packet Block image, which uses the size and length characteristics of the Packet Block to reduce the feature collision of different kinds of traffic, so as to realize the classification of encrypted VPN traffic.

## 3. Method

This paper explores the classification of VPN traffic in TLS tunnels. VPN encapsulation and TLS encryption will bring dual challenges to our traffic classification problem. Most VPN applications add the headers of the VPN client and server the original packets, in order to encapsulate them into new TCP/IP packets. As a result, the header information we obtain is actually the header information of VPN client and server, and the headers of original traffic are hidden. TLS further encrypts the VPN payload and invalidates some methods using payload and plaintext information. The encapsulation of TLS and VPN will have a certain effect on the transmission behavior, but the relative size of packets is not affected by VPN and encryption. Therefore, in recent years, there are many studies on using packet size distribution features to classify VPN traffic.

Qin et al. [20] first proposed the classification of eight applications using PSD features. Shapira et al. [14] used the packet size and packet arrival time to create an image to classify VPN traffic. Both of their studies used packet size distribution features and obtained good classification results. However, the possible feature collision of packet size distribution affects the accuracy of classification. For example, some applications of video and file transfer use the HTTPS protocol, resulting in similar packet size distributions. In this regard, we put forward the concept of Packet Block. Packet Block is the aggregation of a series of consecutive packets in the same direction. Our method takes Packet Block instead of a packet as the basic unit of flow, and uses the length and size of Packet Block to generate images to realize the classification of VPN traffic in a TLS tunnel. Packet Block is not affected by VPN and TLS encryption, and can reflect the data size distribution of the flow and some packet interaction between communication parties. It can be used to classify VPN traffic in TLS tunnels.

### 3.1. Packet Block

Nowadays, most studies on VPN traffic classification regard the packet as the basic unit of flow. Thus, as the flow can be represented as an ordered set of packets, then this model of the flow can be expressed in Figure 1. In the figure, a vertical line represents a packet, the height represents the packet size and the distance between the line represents the time interval between two packets. In order to express the deeper information of the traffic, we put forward the concept of Packet Block, that is, a group of continuous packets in the same direction. We define the number of packets in Packet Block as the length of the Packet Block, and the average size of packets in the packet block as the size of the Packet Block (the upstream traffic is positive and the downstream traffic is negative). [Packet Block's length, Packet Block's size] can be used to indicate a Packet Block. We consider that a flow consists of an ordered Packet Block group, and this model of the flow can be regarded as Figure 2.

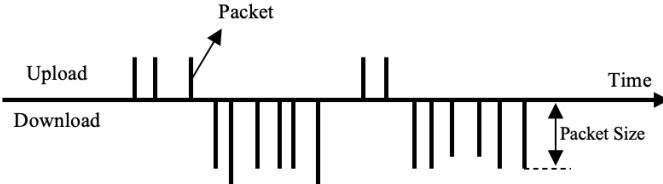

**Figure 1.** Flow model-based packet.

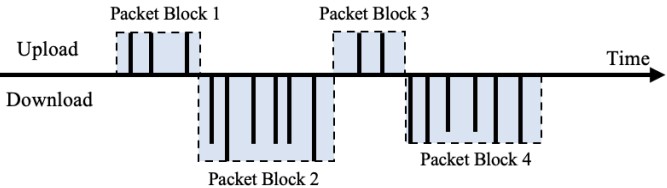

**Figure 2.** Flow model-based Packet Block.

Packet Block feature extraction is simple and is not affected by TLS encryption and VPN encapsulation. Thus, it can be used to classify VPN traffic in TLS tunnels. In the traffic model with the Packet Block, we ignore the time-based feature. This is because time which is greatly affected by the network environment has low robustness, and thus it is of little help to our traffic classification problem. Compared with packet feature, the Packet Block feature can still represent the rule of byte distribution of a flow. In addition, the Packet Block feature can reflect some other deep features, such as the relationship between the upstream and downstream traffic of the communication parties, the data grouping method of the application type and some other deep features. For example, some VPN tunnels may combine multiple packets into one, so it can change the packet size rule of a flow. In a Packet Block, the packet size increases but the packet number reduces. This causes the total size of the Packet Block to adopt a stable value. Therefore, Packet Block feature can deal with VPN and TLS encrypted better than using a single packet.

*3.2. Packet Block Image*

After describing the traffic model in Packet Block, we will create a two-dimensional image of the flow, which is called the Packet Block image. The X axis is the length of the Packet Block and the Y axis is the size of the Packet Block. The value of each pixel represents the number of Packet Blocks with a corresponding length and size. For example, if the value of (4, 1200) is 7, this means that there are seven Packet Blocks with a length and size of 4 and 1200B, respectively. Thus, this image can be regarded as a distribution matrix of the Packet Block length and size. Then, we will use these images as the input of a CNN model. We depict the Packet Block images of five different traffics in OpenVPN dataset, and the images are shown in Figure 3.

It can be seen that several types of traffic show some rules on the Packet Block image. The Packet Block length of VoIP and chat traffic is relatively short, generally within 10. The Packet Block size of chat traffic is smaller than that of VoIP traffic. The Video, FT (file transfer), and browsing traffic have some similarities in the Packet Block size distribution. The downstream Packet Block size is larger than that of the upstream, and the downstream Packet Block size of FT traffic is almost in a straight line. The downstream Packet Block size distribution of video traffic is relatively dispersed, while the downstream Packet Block size distribution of a browsing traffic is the most dispersed. This is because, when a web page is opened, the web page will load a variety of resources, which have different sizes. However, there are some smaller Packet Blocks in video traffic because video traffic is accompanied by a small amount of audio, text and other traffic. When downloading files,

the downstream traffic is relatively pure, which results in the downstream Packet Block size of FT traffic being basically distributed along a straight line near 1400 bytes. In terms of the Packet Block length, that of FT traffic will also be slightly smaller than that of video and browsing traffic.

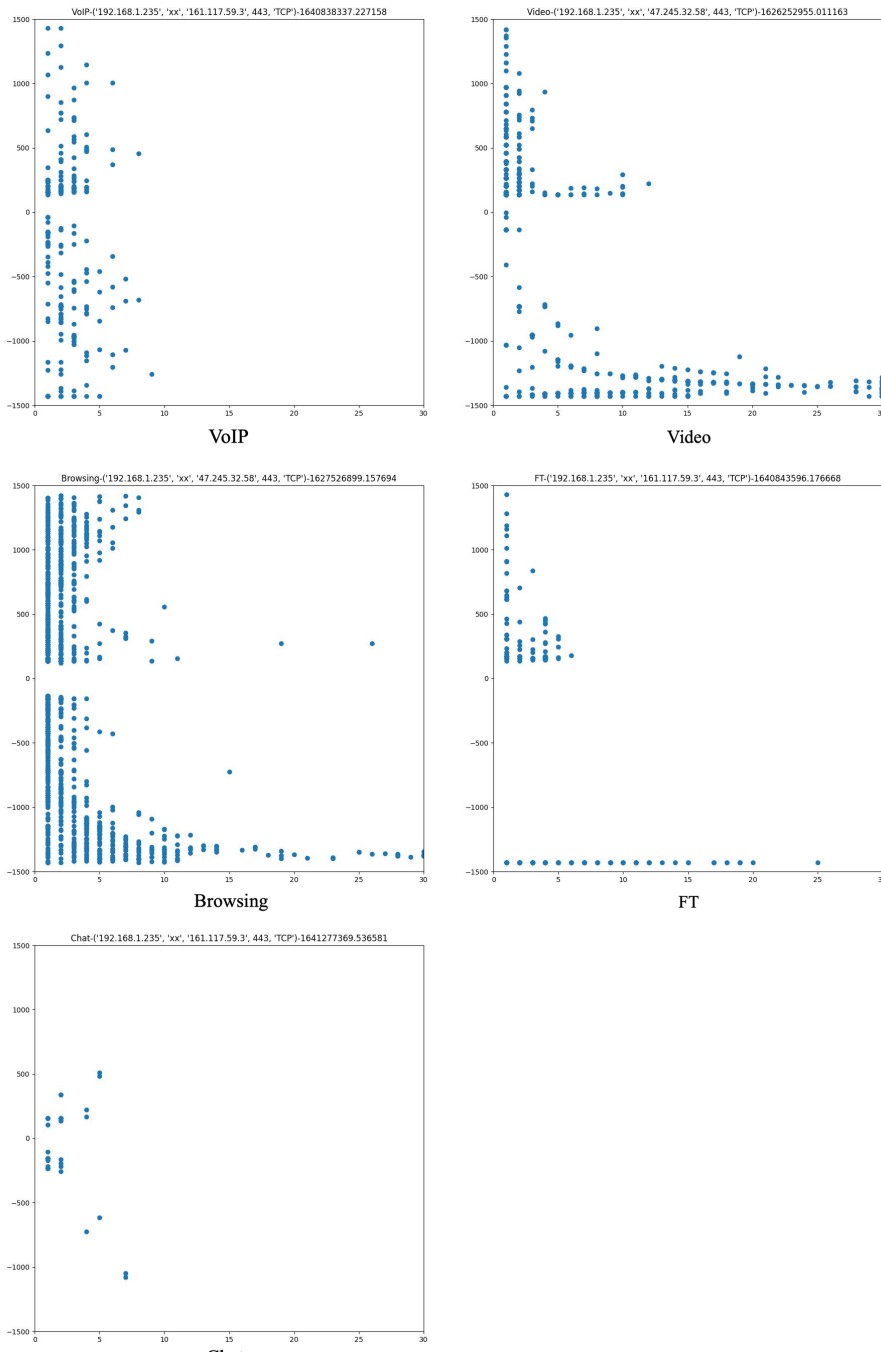

**Figure 3.** Example of Packet Block images.

The X axis is the length of the packet block, with a value ranging from 0 to N. If the value of N is excessively large, the effective part will be compressed on the left side of the image. If the value of N is too small, many packet blocks will be outside the image. Therefore, it is essential to choose an appropriate N value. Observing the packet blocks of different traffics, we initially select the value of N between 10 and 150, and the specific value will be decided through subsequent experiments. The Y axis is the size of the packet

block. The size is less than 1500B, which is the MTU of Ethernet. For the convenience of the calculation, the Y axis value range is $(-1500, 1500]$, with a total of 3000 dimensions. In a proposed classification method, the distribution accurate to 1 byte is unnecessary. We only need the rough distribution of the packet block size. Therefore, we aggregated the Y axis in k-byte units. At this time, the Y axis dimensions M can be seen as $3000/K$. Therefore, the values of the X axis are $\{0, 1, \ldots, N\}$ and those of the Y axis are $\{(-0.5\ MK, -(0.5\ M - 1)\ K], \ldots, (-K, 0], (0, K], (K, 2\ K], \ldots, ((0.5\ M - 1)K, 0.5\ Mk)]\}$.

### 3.3. An Encrypted VPN Traffic Classification Framework Based on CNN Model

This paper proposes an encrypted VPN traffic classification framework based on the CNN model [21] to realize traffic classification. The specific framework is shown in Figure 4. The whole framework is divided into two parts: model generation, verification or classification. The model generation can be divided into three parts: data preprocessing, image generation and model training.

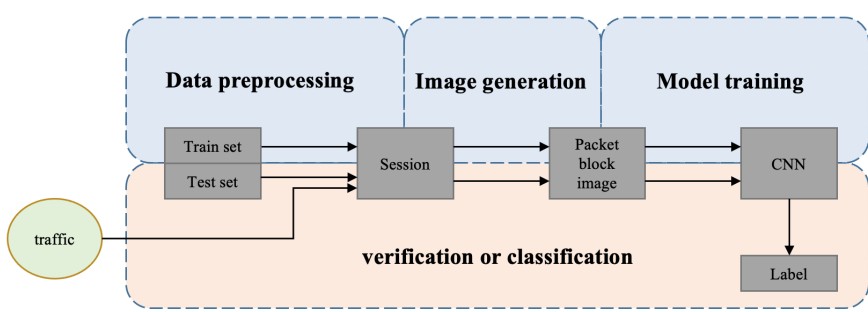

**Figure 4.** Traffic classification framework based on CNN model.

- Data preprocessing: the task of data preprocessing is to extract the flow from dataset files and convert them into data that are easy to process. We use the four-tuple of source IP address, source port, destination IP address and destination port to distinguish different flow. In order to increase the number of datasets and reduce overfitting, we divide the flow into session by time T. A four-tuple plus a start time can determine a session. In this paper, each session is represented by a group of Packet Blocks, and the length and size of each Packet Block are calculated, because our subsequent experiments will only use these two types of information. The labels of each session need to include whether it is VPN traffic, application type and application, such as VPN traffic, video and Tencent. If the number of Packet Blocks in a session is less than 10, we think that the session does not contain valid information and round it off. Finally, the format of each session is shown in Figure 5.

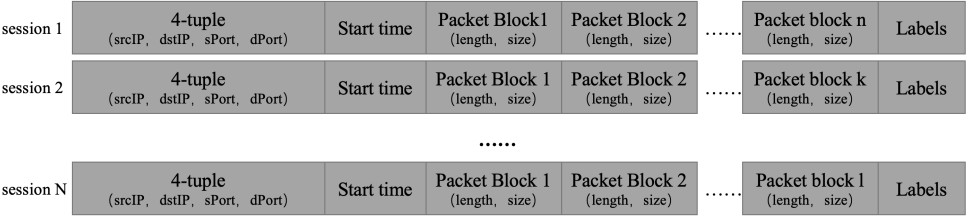

**Figure 5.** Example of sessions.

- Image generation: In this step, we generate the Packet Block image by using the session obtained by data preprocessing. In this process, we need to determine two parameters: the aggregation degree K of the Packet Block size and the upper limit N of Packet Block length. If the length of the Packet Block exceeds the upper limit N, the length of the packet block will be regarded as N. The final Packet Block image is an M*N matrix (M = 3000/K).

- Model training: This step mainly trains the CNN model through the images set obtained from the image generation in the previous step. There are two reasons why we choose the CNN model. First, the deep learning model can reduce the dependence on handcrafted features. Secondly, local features can be extracted by local connections of the convolution layer in the CNN model, which is compatible with our Packet Block feature.
- Verification or classification: In the verification or classification step, the test dataset or traffic to be detected can be preprocessed and converted into an image, then imported into the classification model to calculate the final labels.

## 4. Experiments and Results

### 4.1. Dataset

The ISCX-tor [22] dataset and ISCX-VPN [18] dataset are widely used by researchers in traffic classification research. However, the ISCX-VPN dataset is not completely applicable to this study. On the one hand, the volume of some types of traffic in ISCX-VPN dataset is small, such as chat and email. If the dataset is directly applied to the experiment, the imbalance may have some influence on the final result. On the other hand, all types of traffic in the ISCX-VPN dataset are not TLS tunnel traffic, which is not completely consistent with our research scenario. Therefore, the ISCX-Tor dataset and the OpenVPN dataset captured by ourselves are used in this paper. The ISCX-Tor dataset consists of the tor traffic and non-tor traffic of seven application types. Five of them are selected in our study: VoIP, video, file transfer, chat and browsing. The OpenVPN dataset we captured also includes these five application types. The application types and specific applications of ISCX-Tor dataset and VPN dataset captured by us are shown in Table 1.

**Table 1.** Protocols and applications for each traffic category.

|  | ISCX-Tor | OpenVPN |
|---|---|---|
| VoIP | Hangouts, Facebook, Skype | QQ, WeChat, Skype |
| Video | Vimeo, YouTube | Tencent, Aiqiyi, YouTube |
| FT | FTP, SFTP, Skype | SFTP, FTPS, SCP |
| Chat | Hangouts, Facebook, AIM, Skype, ICQ | QQ, WeChat, Skype |
| Browsing | Firefox, Chrome | Firefox, Chrome, Safari |

In order to create a complete and representative dataset, this study set up a VPN environment in the laboratory by referring to the capture method of the ISCX-VPN [18] dataset to capture the VPN proxy tunnel traffic encrypted by the TLS of different types and applications. Referring to studies on traffic classification, most of them use traffic from five application types, namely video, VoIP, file transfer, chat and browsing, to classify traffic. Therefore, the dataset of this study includes the traffic of these five application types. For each type of traffic, we captured a regular session and a session over a VPN tunnel, so there are 10 traffic categories. All application types and specific applications contained in the OpenVPN dataset captured by us are shown in Table 1.

The lab environment is shown in Figure 6, where a VPN gateway is set up between the PC and the campus network gateway, where the OpenVPN client and Stunnel are deployed. The ENP3s0 port of the VPN gateway is connected to the PC, and the ENP1s0 port is connected to the external network through the campus network. With this configuration, we run the TCP-dump on the VPN gateway and capture a pair of .pcap files on enp3s0 and ENp1s0 ports, marking non-VPN traffic and VPN traffic, respectively.

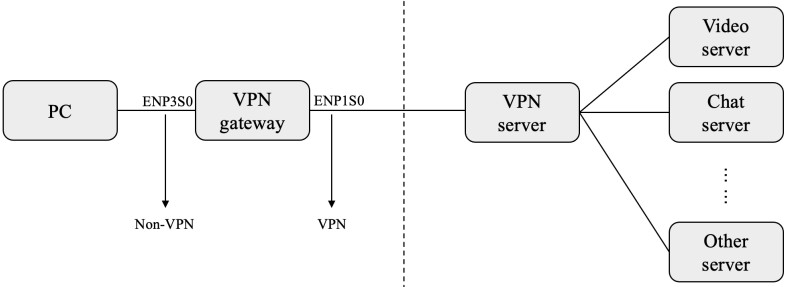

**Figure 6.** Network environment of the captured dataset.

The source IP, destination IP, source port and destination port of the VPN traffic we capture are exactly the same, which makes it difficult for us to distinguish them. Therefore, we control the PC to run only one program in a time period, and all traffic tags in this time period are the program. In addition, we also need to observe the non-VPN traffic captured by the ENP3s0 port to ensure that the noise traffic is within an acceptable range. Otherwise, we will discard the .pcap file. Below, we give a detailed description of different types of traffic generation:

- Video: We chose three common video apps, namely Tencent, Aiqiyi and YouTube. Each application is run manually when capturing traffic.
- VoIP: All traffic generated by voice applications. We chose QQ, WeChat and Skype to run each application manually when capturing traffic.
- File transfer: Mainly refers to the traffic related to file upload or download. File transfer traffic is transmitted over different protocols. Therefore, FTPS, SFTP, and SCP are selected. Each application is run manually when capturing traffic.
- Chat: In this category, we select QQ, WeChat and Skype. Each application was run manually when capturing the traffic or using an automated chatbot to automatically generate chat messages.
- Browsing: We used Firefox, Chrome and Safari browsers, and used Python scripts to automatically visit the TOP100 websites on Alexa when generating traffic. Each website was visited five times with an interval of 5 s between two visits.

For each application of each traffic, we captured the .pcap file of 3 h. Finally, 29.47 g VPN traffic and 25.39 g non-VPN traffic are captured. VPN traffic and non-VPN traffic had five application types: video, VoIP, file transfer, chat and browsing. Each type has three applications or protocols. See Table 1 for details.

In the data preprocessing of this experiment, in order to increase the number of samples in the dataset and reduce overfitting, the flow is divided into 30 s' sessions. Table 2 shows the number of sessions of various application types in the ISCX-tor dataset and the captured OpenVPN dataset. It can be seen that the number of sessions in the OpenVPN dataset is greater than that in the ISCX-tor dataset, and the number of chat type sessions is greatly increased compared with that in the ISCX-tor dataset, thus ensuring the balance of the dataset.

**Table 2.** The number of sessions in ISCX-Tor and OpneVPN.

|  | **ISCX-Tor** | **OpenVPN** |
|---|---|---|
| VoIP | 755 | 955 |
| Video | 302 | 2114 |
| FT | 302 | 1520 |
| Chat | 51 | 451 |
| Browsing | 502 | 634 |

*4.2. Measurement*

In this study, we use the accuracy (*ACC*) as the measurement. Accuracy is not only one of the most common measurements in deep learning, but also the main measurement adopted by many flow classification methods such as FlowPic. The definition of accuracy in this experiment is as follows.

$$ACC = \frac{\sum_{i \in A} TP_i}{\sum_{i \in A} (TP_i + FP_i)}. \tag{1}$$

where $A$ = {VoIP, Video, FT, Chat, Browsing}; the true positive (*TP*) and false positive (*FP*), respectively, represent the number of positive samples correctly classified and the number of negative samples incorrectly classified as positive samples; and $TP_i$ and $FP_i$ represent the number of true positives and false positives of category *i*, respectively. The advantage of *ACC* is intuitive. *ACC* directly represents the proportion of correctly predicted samples. However, there is also a problem with the *ACC* index. If there is a large number of samples in one class in the dataset, even if the classifier has a poor classification effect on other classes of samples, it can still maintain a high *ACC* in the end.

In order to avoid this situation, in addition to *ACC*, we also used the confusion matrix to better observe the multi classification problem. In the confusion matrix, each row represents the real label and each column represents the predicted label. Diagonals indicate the probability of the correct prediction for each category. Precision, recall and F1-score are also used. In this experiment, they are defined as follows:

$$Precision = \frac{TP}{TP + FP}. \tag{2}$$

$$Recall = \frac{TP}{TP + FN}. \tag{3}$$

$$F1\text{-}score = \frac{2 * Precision * Recall}{Precision + Recall}. \tag{4}$$

*4.3. CNN Model*

Traffic classification in this paper adopts the classification framework based on the aforementioned CNN model, and the parameter settings of the CNN model are shown in Table 3. Our CNN model is divided into seven layers, and the input is a matrix of (60, 60). Parameter Settings for seven layers are as follows:

**Table 3.** The main parameters of CNN model.

| Layer | Layer Name | Filter | Stride | Output |
|-------|-----------|--------|--------|--------|
| 1 | Conv1D_1 | 5*(6,6) | 1 | (60,60,5) |
| 2 | MaxPooling1D_1 | (3,3) | 3 | (20,20,5) |
| 3 | Conv1D_2 | 10*(5,5) | 1 | (20,20,10) |
| 4 | MaxPooling1D_2 | (2,2) | 2 | (10,10,10) |
| - | Dropout | - | - | (10,10,10) |
| 5 | Flatten | - | - | 1000 |
| 6 | Dense | - | - | 64 |
| - | Dropout | - | - | 64 |
| 7 | Softmax | - | - | 5 |

- Conv1D_1: the first layer is the first convolutional layer, which has five filters with a convolution kernel size of 6 and a step size of 1. The filling mode is set to "SAME", so the output vector size is (60,60,5). There are 185 trainable parameters in this layer.
- MaxPooling1D_1: the second layer is the maximum pooling layer and the output size is (20,20,10). Convolution layer plus maximum pooling layer is sometimes referred to as a convolution unit. The model selected in this experiment has two convolution units.
- Conv1D_2: The third layer is the second convolutional layer, which has 10 filters with a convolution kernel size of 5 and step size of 1. The filling mode is set to "SAME", so the output vector size is (10,10,10). There are 1260 trainable parameters in this layer.
- MaxPooling1D_2: the fourth layer is the maximum pooling layer and the output size is (10,10,10).
- Flatten: the fifth layer is the flat layer, which transforms the feature mapping of (10,10,10) into a one-dimensional layer with a size of 1000.
- Dense: The sixth layer is a 64 fully connected layer with 64,000 trainable parameters.
- Softmax: The seventh layer is the output layer, whose size depends on the number of categories of classification. In traffic classification problems of five application types, the size of this layer is 5. The loss function of the model adopts the cross-entropy loss function. For the optimization process, we use the Adam optimizer and use the default super parameters provided by Kingma et al. [23]. We use keras as the back-end to build and run our network, which runs a total of 40 epochs.

### 4.4. Packet Block Image Size Selection Experiment

In our experiment, the aggregation degree K of the Packet Block size and the upper limit N of Packet Block length affect the size of the images. The size of the images can be calculated as M*N (M = 3000/K). To choose the appropriate value of M and N, we designed two groups of experiments on OpenVPN dataset. The first group set M = 30 to observe the impact of different N values on the classification accuracy. The second group set N = 60 to observe the impact of different M values on the classification accuracy. We conducted five VPN traffic classification experiments on two groups to choose the value of M and N.

Table 4 shows the experimental results of traffic classification under different N values when M = 30. It can be seen that the classification accuracy increases with the increase in N at the beginning. When N increases to 60, the classification accuracy remains almost stable. This is because the packet block length of most traffic is less than 60. The continuous growth of N does not provide more available information for the packet block image. In addition, when N = 60, the time of running each epoch is only 1 s longer than when N = 10. Therefore, in subsequent experiments, we will choose the value of N as 60.

Table 5 shows the experimental results of traffic classification under different M values when N = 60. It can be seen that M ranges from 30 to 600, and the value of classification accuracy changes very little. This is because we only need to understand the approximate byte distribution of traffic, rather than accurate to each byte, for traffic classification problems of different application types. Therefore, the 30-dimensional packet block size feature that aggregates 100 bytes can achieve a good accuracy. In terms of running time, the time taken by the model to run an epoch will decrease significantly after M becomes smaller. Therefore, we choose the value of M as 60 in the subsequent experiments.

### 4.5. Traffic Classification Experiment Based on Packet Block Image

We will select the parameters obtained above, i.e., m = 60, n = 60, and conduct traffic classification experiments for five application types of traffic on our captured OpenVPN dataset and public ISCX-tor dataset. The classifier adopts the CNN model described above. The ACC, precision, recall and F1-score of the final traffic classification problem are shown in Table 6. The classification ACC of the packet block image method on the OpenVPN dataset is 97.20%, and the classification ACC on the ISCX-tor dataset is 93.31%. Furthermore, the F1-scores in the two datasets are 96.70% and 89.28%.

**Table 4.** Results under different N when M = 30.

| Length N | Train ACC | Test ACC | Time per Epoch |
|----------|-----------|----------|----------------|
| 10 | 88.30% | 92.35% | 1 s |
| 20 | 93.04% | 94.98% | 1 s |
| 30 | 94.09% | 95.48% | 1 s |
| 40 | 94.75% | 95.15% | 2 s |
| 50 | 95.44% | 96.13% | 2 s |
| 60 | 96.15% | 96.63% | 2 s |
| 70 | 95.74% | 95.64% | 2 s |
| 80 | 96.24% | 96.30% | 3 s |
| 100 | 96.17% | 96.46% | 9 s |
| 120 | 96.13% | 96.55% | 11 s |
| 150 | 97.00% | 96.63% | 20 s |

**Table 5.** Results of different M when N = 60.

| Size M | Train ACC | Test ACC | Time per Epoch |
|--------|-----------|----------|----------------|
| 30 | 96.15% | 96.63% | 2 s |
| 60 | 96.34% | 97.20% | 7 s |
| 300 | 97.66% | 96.55% | 30 s |
| 600 | 98.24% | 96.79% | 107 s |

**Table 6.** Results in OpenVPN and ISCX-Tor.

| | OpenVPN | | | | ISCX-Tor | | | |
|---|------|-----------|--------|--------|------|-----------|--------|--------|
| | ACC | Precision | Recall | F1 | ACC | Precision | Recall | F1 |
| VoIP | - | 0.9904 | 0.9858 | 0.9881 | - | 1 | 0.9934 | 09967 |
| Video | - | 0.9790 | 0.9768 | 0.9779 | - | 0.9778 | 0.7719 | 0.8627 |
| FT | - | 0.9809 | 0.9961 | 0.9885 | - | 1 | 0.9655 | 0.9825 |
| Chat | - | 0.9487 | 1 | 0.9737 | - | 1 | 0.5714 | 0.7273 |
| Browsing | - | 0.9496 | 0.8626 | 0.9040 | - | 0.8173 | 0.9844 | 0.8947 |
| Average | 0.9720 | 0.9698 | 0.9642 | 0.9670 | 0.9331 | 0.9590 | 0.8581 | 0.8928 |

OpenVPN dataset is 97.20%, and the classification ACC on the ISCX-tor dataset is 93.31%. It can be seen that the ACC of our Packet Block image method for five traffic classification problems under the ISCX-Tor dataset is significantly higher than that of FlowPic method (67.8%). Even though FlowPic balances ISCX-Tor dataset, our ACC is still higher than their 86.9%. Furthermore, under the OpenVPN dataset captured by ourselves, the ACC of classification can reach 97.20%. As for the speed, it takes approximately 5–10 min for FlowPic to run an epoch, while the Packet Block image method only needs 3 s to run an epoch because the dimension of features is greatly reduced. Therefore, the processing speed of the Packet Block image method is much faster than the FlowPic method.

In order to observe the multi-classification results more clearly, we give the confusion matrix of the Packet Block image method under the OpenVPN dataset and ISCX-tor dataset, as shown in Figures 7 and 8. In the OpenVPN dataset, the number of correctly identified samples exceeds 97% in the four types of traffic: VoIP, video, FT and chat. However,

the classifier has a poor recognition effect on browsing, and only 86.26% of the browsing traffic is correctly recognized, and approximately 14% of browsing traffic is recognized as video, VoIP and FT traffic. Under the ISCX-tor dataset, we can first see that the identification effect of the chat traffic is not good, which is within our expectations. This is because various types of traffic in the ISCX-tor dataset are unbalanced, and the number of chat traffic is much smaller than that of other types of traffic. In addition, the classifier has the worst recognition ability for video, and 22.81% of video traffic is recognized as browsing traffic. However, it is worth noting that all misclassified traffic is related to browsing traffic. Some browsing traffic is identified as other types, and some other types are identified as browsing. If the browsing traffic is ignored, the classification accuracy of the other four traffic types will be greatly improved. It can be seen that regardless of whether the OpenVPN dataset or ISCX-tor dataset is used, the browsing traffic will have a great influence on the traffic classification. This is because, when we label the dataset, the browsing traffic overlaps with the other four types of traffic. For example, when we visit the main page of some websites, a video will be played automatically. At this time, the browsing traffic is actually also a video traffic. This leads to the limitation of browsing traffic in traffic classification.

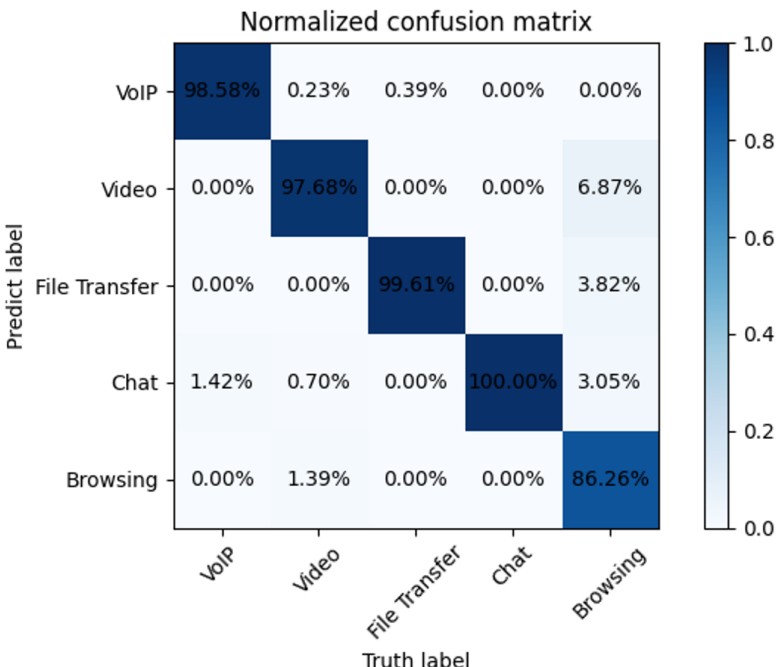

**Figure 7.** Confusion matrix in an OpenVPN dataset.

*4.6. Comparison with Related Methods*

We compare the proposed method with some traditional machine learning methods and recent deep learning methods in the ISCX-Tor dataset. The results are shown in Table 7. We can see that the methods based on deep learning have an apparent improvement rather than traditional ML methods. Furthermore, our proposed method achieves the best accuracy and F1-score in recent deep learning works. In these methods, the results of PSD and end-to-end methods are reproduced by us in the ISCX-Tor dataset, because they use different datasets. It can be seen that the accuracy of the end-to-end method is only above 20%. This is because payload-based methods have no ability to deal with encrypted traffic. Overall, our proposed approach can deal with VPN traffic classification using TLS encryption, and have a good improvement over the recent methods.

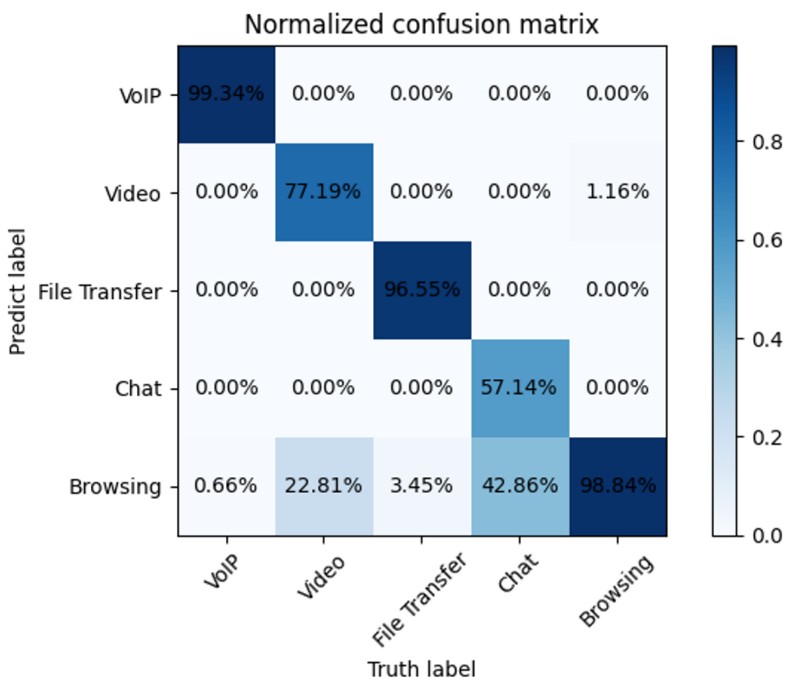

**Figure 8.** Confusion matrix in ISCX-tor dataset.

**Table 7.** Results compared with related methods.

|  |  | ACC | Pr | Re | F1 |
|---|---|---|---|---|---|
| Packet Block image |  | 0.9331 | 0.9590 | 0.8581 | 0.8928 |
| FlowPIC |  | 0.8964 | - | - | 0.6493 |
| Time-related |  | 0.8330 | 0.8410 | 0.8310 | 0.8360 |
| PSD |  | 0.8089 | 0.8995 | 0.8561 | 0.8820 |
| End to end |  | 0.2032 | 0.2016 | 0.1920 | 0.1967 |
| Traditional ML methods | DT | 0.7755 | - | - | 0.6331 |
|  | KNN | 0.6203 | - | - | 0.4434 |
|  | SVM | 0.5169 | - | - | 0.2298 |

## 5. Conclusions

This paper aims to solve the classification problem of VPN traffic in the TLS tunnel. The encryption of the TLS tunnel and the encapsulation of VPN are two of the main difficulties in the traffic classification problem. In this paper, we propose an encrypted VPN traffic classification method based on Packet Block image. In this method, we represent the flow with a sequence of Packet Blocks, and then generate a Packet Block image of it. Actually, the Packet Block image represents the distribution of packet blocks in VPN traffic. These images are then imported into CNN for learning. Packet Block images can extract deep features besides the packet size distribution, which can greatly avoid feature collision, so as to ensure the classification result. The public authoritative ISCX-tor dataset and our OpenVPN dataset are selected to verify our method, and the traffic classification experiments of five application types are carried out on the two datasets, respectively. Finally, the Packet Block image method has a classification ACC of 93.31% under the ISCX-tor dataset and 97.20% under the OpenVPN dataset we captured, which a has higher accuracy than the FlowPic method. Because the size of the method image of the Packet Block image is smaller than that of FlowPic, the processing speed of the model is much

higher than that of FlowPic. Therefore, it was verified that our Packet Block image method can be used in the classification problems of VPN traffic in the TLS tunnel.

However, if many different flows are mixed, the effectiveness of our approach will be diminished. Almost all methods run their experiments using a single-flow dataset. However, most real-world traffics are mixed, such as watching a video when downloading some files. Thus, the identification and division of mixed VPN traffic is an important research in the future.

**Author Contributions:** Conceptualization, S.Z. and W.S.; Methodology, S.Z. and W.S.; Software, W.S.; Validation, Y.Z., J.L. and C.S.; Resources, Z.S.; Data Curation, W.S. and Y.Z.; Writing—Original Draft Preparation, W.S.; Writing—Review and Editing, S.Z.; Project Administration, S.Z. All authors have read and agreed to the published version of the manuscript.

**Funding:** This research received no external funding.

**Data Availability Statement:** Publicly available datasets were analyzed in this study. The Tor-nonTor datasets can be found here: https://www.unb.ca/cic/datasets/tor.html. And the data we created can be found here: https://github.com/spirit19970507/OpenVPN (accessed on 14 October 2022).

**Conflicts of Interest:** The authors declare no conflict of interest.

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
