# Peer review of "A Deep Learning-Based Encrypted VPN Traffic Classification Method Using Packet Block Image"

_electronics, doi:10.3390/electronics12010115_

Round 1

Reviewer 1 Report

In recent years, the using of VPN and TLS encryption makes network traffic classification face new challenges. This paper proposes a deep learning-based encryption traffic classification method, which uses convolutional neural networks to process Packet Block features extracted from network traffic to identify the application type of network traffic. The Packet Block feature improves the feature conflict problem of feature sequences with a single packet as the basic unit in different protocols and traffic classes. However, the following problems still need to be solved:

1.     This article is not clear about the problem that needs to be solved. For example, it can be seen from the experimental part that the article is mainly an improvement on the existing work [14], but lacks a clear description in the introduction part.

2.     There are few researches on existing related work in the article, which cannot explain the latest research progress.

3.     What is the data source for the Packet Block image shown in Section 3.2? Just showing the Packet Block image doesn't really describe the advantages of using Packet Block versus a single packet

4.     The description of generating Packet Block images from sessions is unclear in Section 3.3. In M = 3000/K, how exactly are 3000 and K obtained? How is the M*N matrix constructed?

5.     In the experimental part, the evaluation index only uses the accuracy and cannot reflect the real results. And there is a lack of comparative experiments with existing work.

Reviewer 2 Report

The authors proposed a paper titled “A Deep Learning-Based Encrypted VPN Traffic Classification Method Using Packet Block Image”. A traffic classification method based on deep learning is provided in this paper. The research topic is very important; the following minor modifications are suggested to improve the quality of the paper:

- Authors should summarize the literature review in some form of comparison table to draw conclusions.

-     Experimental Results section should be updated by adding more details about the comparison between the proposed approach and the other approaches in the literature.

Author Response

Thank you for your comments on my paper. I have read it carefully and revised my paper.

Point:

Authors should summarize the literature review in some form of comparison table to draw conclusions.

Experimental Results section should be updated by adding more details about the comparison between the proposed approach and the other approaches in the literature.

Response:

In Section 4.5, we redrew the Table 6. We added more experimental measurement such as: precision, recall, F1-score.

We added a section(Section 4.6, line 431-441) to compare our method with others. And we gave the Table 7. In this table, we compared our method with 7 methods in accuracy, precision, recall and F1-score.

Reviewer 3 Report

This paper deals with the problem of traffic classification when there exist VPN encapsulation and TLS encryption. The authors propose to represent network flows as sequences of Packet BLocks, then generate Packet Block images and input these images into CNN model for traning. The proposed method shows high classification accuracy for 5 flows: VoIP, video, FT, chat, browsing.

This research provides an interesting idea, and in general, the paper was well presented. There are only a few concerns:

1.
1.1  At some devices (e.g., gateway of campus network) packets of many flows are mixed, so 1 Packet Block may contain packets from several flows or packets with different types of traffic. So where and how we can apply the proposed solution?

1.2 Also about the issue of mixed flows, the authors wrote "control the PC to run only one program in a time period" or "Run each application manually when capturing traffic", but in reality, several programs may run in parallelin in one machine?

1.3 Similarly, as the authors wrote "For example, when we visit the main page of some websites, a video will be played automatically", 1 flow may contain several types of traffic. There are some other examples: file sending inside a chat session, file downloading/uploading when browsing, video playing in a chat, video playing in a VoIP. This should be a critical issue in future works?

2. If the tunneling process also combines packets and makes the packet size become less predictable, there will be impact to the proposed method?

Author Response

Thank you for your comments on my paper. I have read it carefully and revised my paper.

Point 1:

1.1  At some devices (e.g., gateway of campus network) packets of many flows are mixed, so 1 Packet Block may contain packets from several flows or packets with different types of traffic. So where and how we can apply the proposed solution?
1.2 Also about the issue of mixed flows, the authors wrote "control the PC to run only one program in a time period" or "Run each application manually when capturing traffic", but in reality, several programs may run in parallelin in one machine?
1.3 Similarly, as the authors wrote "For example, when we visit the main page of some websites, a video will be played automatically", 1 flow may contain several types of traffic. There are some other examples: file sending inside a chat session, file downloading/uploading when browsing, video playing in a chat, video playing in a VoIP. This should be a critical issue in future works?

Response 1:

Our approach is to deal with relatively single traffic, the effect for mixed traffic will be reduced. If we watch video when chatting, the traffic we get is a mixed traffic. But there is always a period of time when video is the main traffic. Our approach is for this period of time. And identification and division of the mixed traffic is a critical issue in future works.

We gave an explanation in Section 5, paragraph 2, line 460-464.

Point 2:

If the tunneling process also combines packets and makes the packet size become less predictable, there will be impact to the proposed method?

Response 2:

If the tunnel combines two packets into one, their packet size will increase, but the length of packet block (the number of packets in the packet block) will decrease, then the total size of a packet block will be a relatively stable value. Therefore our method can deal with this case. We add an explanation in Section 3.1, paragraph 2, line 203-207